# Amino acid, sugar, phenolic, and terpenoid profiles are capable of distinguishing *Citrus tristeza virus* infection status in citrus cultivars: Grapefruit, lemon, mandarin, and sweet orange

**Christopher M. Wallis**[1]*, **Zachary Gorman**[1], **Rachel Rattner**[1], **Subhas Hajeri**[2], **Raymond Yokomi**[1]

**1** Crop Diseases, Pests and Genetics Research Unit, United States Department of Agriculture—Agricultural Research Service San Joaquin Valley Agricultural Sciences Center, Parlier, California, United States of America, **2** Citrus Pest Detection Program, Central California Tristeza Eradication Agency, Tulare, California, United States of America

* christopher.wallis@ars.usda.gov

**Data Availability Statement:** All data will be available at Ag Data Commons, part of the United States National Agricultural Library, at: https://doi.

## Abstract

*Citrus tristeza virus* (CTV) is the most severe viral disease for citrus production. Many strains of CTV have been characterized and their symptomology widely varies, ranging from asymptomatic or mild infections to severe symptomology that results in substantial yield loss or host death. The capacity of the different CTV strains to affect the biochemistry of different citrus species has remained largely unstudied, despite that associated metabolomic shifts would be relevant toward symptom development. Thus, amino acid, sugar, phenolic, and terpenoid levels were assessed in leaves of healthy and CTV-infected grapefruit, lemon, mandarin, and two different sweet orange cultivars. Both mild [VT-negative (VT-)] and severe [VT-positive (VT+)] CTV genotype strains were utilized. When looking at overall totals of these metabolite classes, only amino acid levels were significantly increased by infection of citrus with severe CTV strains, relative to mild CTV strains or healthy plants. No significant trends of CTV infection on summed amounts of all sugar, phenolic, or terpenoid compounds were observed. However, individual compound levels were affected by CTV infections. Subsequent canonical discriminant analysis (CDA) that utilized profiles of individual amino acids, terpenoids, or phenolics successfully distinguished leaf samples to specific citrus varieties and identified infection status with good accuracy. Collectively, this study reveals biochemical patterns associated with severity of CTV infections that can potentially be utilized to help identify in-field CTV infections of economic relevance.

org/10.15482/USDA.ADC/1524468 (878beadb-38bc-44be-88b7-441df49bacfb)."

**Funding:** RY received a grant (Project 5300-185) from the Citrus Research Board/ Tulare Pest Control District Board (https://citrusresearch.org/) to perform this research. CMW and RY also supported this research with funds allocated to the United States Department of Agriculture-Agricultural Research Service (https://www.ars.usda.gov/), Projects 2034-22000-012-00D and 2034-22000-013-00D. The funders had no role in study design, data collection and analysis, decision to publish, or preparation of the manuscript.

**Competing interests:** The authors have declared that no competing interests exist.

## Introduction

*Citrus tristeza virus* (CTV) (family *Closteroviridae*, genus *Closterovirus*), the major viral pathogen threat to citrus production [1], is comprised of a complex of distinct strains [2, 3]. CTV is a foregut-born, semi-persistent virus spread by aphid vectors including *Aphis gossypii* and *Toxoptera citricida* [4].

Symptomology of CTV is driven by the viral strain, citrus cultivar, and host scion/rootstock combination. Therefore, the effects of CTV infection can vary widely in terms of severity [2, 3]. Some infections result in severe symptoms that greatly impact yield, which can include quick decline (often caused by girdling at a sour orange rootstock), stem pitting in trunks and branches, poor fruit quality and production, and more. Alternatively, some CTV strain/cultivar interactions result in mild or symptomless infections, which generally occur in tolerant cultivars [5]. This variability in symptomology makes effective management of CTV difficult where neighboring citrus cultivars are different and necessitates the identification of specific CTV strains that result in severe stunting, stem pitting, and brittle branches. CTV monitoring and rogueing has been utilized in certain areas for CTV management [6]. However, non-discriminatory practices may remove both citrus infected with strains of CTV that are virulent and have yield-impacting symptoms as well as citrus infected with asymptomatic CTV strains that will not affect yield [4]. Thus, rogueing of only specific CTV strains that result in yield-reducing symptoms is economically advantageous.

For the purpose of this study, there are two defined strain types of CTV considered to cause greater severity than others, called non-VT MCA13 [7, 8] and VT MCA13 strains [9, 10]. Both of these have been specifically targeted in surveys in the past due to the chance that they could cause severe disease symptoms [11–13]. However, studies have observed that among these strains there can be great variability in the severity caused, including no clear differences between some of these strains and those considered milder [3, 14, 15]. Among these categories, additional CTV strains of concern emerge including resistance-breaking (RB) strains that replicate in trifoliate orange rootstocks once thought to be immune to CTV [16] and other strains that are not present in California [3]. Thus, a need exists for further bio-characterization of different strains on different citrus cultivars [14] including examinations of the same strains across different citrus species and focusing on underlying changes in host metabolism caused by different strains.

Though extensive characterization of different CTV strain/cultivar combinations has been performed [5], there has been few investigations of the biochemical mechanisms behind different CTV symptomologies. CTV, like other viruses, induces physiological changes within its hosts as the result of the infection process. Some of these changes could enhance viral replication or survival in host cells, such as creating new "sinks" for primary metabolites (amino acids and simple sugars) to increase raw materials required for viral reproduction in infected tissues [17]. Other changes could involve alteration of host defense or stress responses [18, 19]. For example, altered phenolic or volatile compound production could render tissues more suitable for vectoring insects, including reduction of volatile terpenoids involved in defense against CTV vectors [20]. Some additional changes in host physiology could create more stable environment for virion populations, such as shifts to osmoregulatory or antioxidant compounds in infected tissues to limit stresses that otherwise might be difficult to overcome [21]. Some of these physiological changes could represent generic host plant responses to viruses, but many are likely to be specific to pathosystems of different host species and viral pathogens or strains.

Only a few studies have examined the effects of CTV on host chemistry [20, 22–24]. These were generally performed to develop diagnostic techniques and not to clarify symptom

development and were almost entirely focused on volatile terpenoids. Cheung et al. [22] observed that a multivariate statistical approach could distinguish volatile emissions of CTV-infected trees from those of healthy trees, and that infected trees had less monoterpenoid production than healthy plants. In another study, CTV-susceptible and resistant varieties of citrus were analyzed, with resistant varieties exhibiting greater levels of terpenoids than susceptible varieties [20]. Terpenoid production is not only dependent on host scions, as a recent study showed that rootstocks can also affect the amount of terpenoids in the scion [24]. Notably, infected plants were not examined by either Guarino et al. [20] or Albrecht et al. [24], so it remains unclear how host volatiles would have changed in those studies following CTV infection.

This study aimed to better characterize host physiological changes in different citrus cultivars following infection by different CTV genotypes possessing different capacities to cause disease. Furthermore, phenolics, terpenoids, amino acids, and free sugars were profiled in order to better elucidate how these metabolite classes are modulated by infection with mild or severe CTV strains. Metabolite profiles of healthy or infected plants, from each host plant cultivar, were then used to see if the citrus species and/or CTV infection status of samples could be correctly identified. Altogether, results from this study improve understanding about virus-plant interactions and highlight host physiological changes that may be favorable for viral survival and spread.

## Materials and methods

### Plant materials, viral inoculations, and CTV titer experiments

Six citrus cultivars propagated on Carrizo rootstock were obtained from a commercial citrus nursery using a certified clean (pathogen-free) budwood and transplanted into one-gallon (3.8 L) pots with UC citrus soil mix [25] and established in a greenhouse. The citrus cultivars included Lisbon lemon (*Citrus limon* L. Burm.f.), Minneola (*Citrus × tangelo*), W. Murcott (*C. reticulata* Blanco), Oro Blanco grapefruit (*C. paradisi* Macfadyen), Midknight Valencia and Washington Navel sweet oranges (*C. sinensis* (L.) Osbeck) for a total of six cultivars. After six months, plants were graft inoculated [26] with five different genotypes of CTV including T30 (one), T36 (one), S1 (one) RB (one) and VT (five) strains for a total of nine CTV treatments and a mock inoculated healthy control. These strains were isolated and characterized from commercial citrus orchards in central California except SY568 which was isolated from citrus in Riverside, California. For purposes of this study, the virus strains and its phenotype were simplified as non-VT = "mild" strains or VT-positive = "severe" strains (Fig 1).

CTV infection status of the citrus propagations was determined by double antibody sandwich indirect (DAS-I) enzyme linked immunosorbent assay (ELISA) [27] with a polyclonal CTV antiserum [28] and MCA13 [7]. Once CTV systemic infection was established in all plants, the plants were moved to a screenhouse covered with 50 mesh (hole size 0.78 x 0.25 mm) anti-insect (whiteflies, aphids, psyllids, leaf-miners, and thrips) net and planted in the ground in a randomized split plot design with three reps. The screenhouse was 26 feet (7.9 m) wide and 120 feet (36.6 m) in length and tree spacing was 3 feet (0.9 m) between trees and 6 feet (1.8 m) between rows. The screenhouse as equipped with a screened entrance vestibule but no air circulation or controlled temperature systems. During summer, a shade cloth was added on the west exposure of the screenhouse to help reduce late afternoon heat load. To facilitate symptom evaluation, the three reps were planted next to each other in the same row and each row constituted a different cultivar. Trees in the screenhouse were irrigated by microjet sprinkler. To monitor CTV titer in plants in the screenhouse, Reverse Transcription

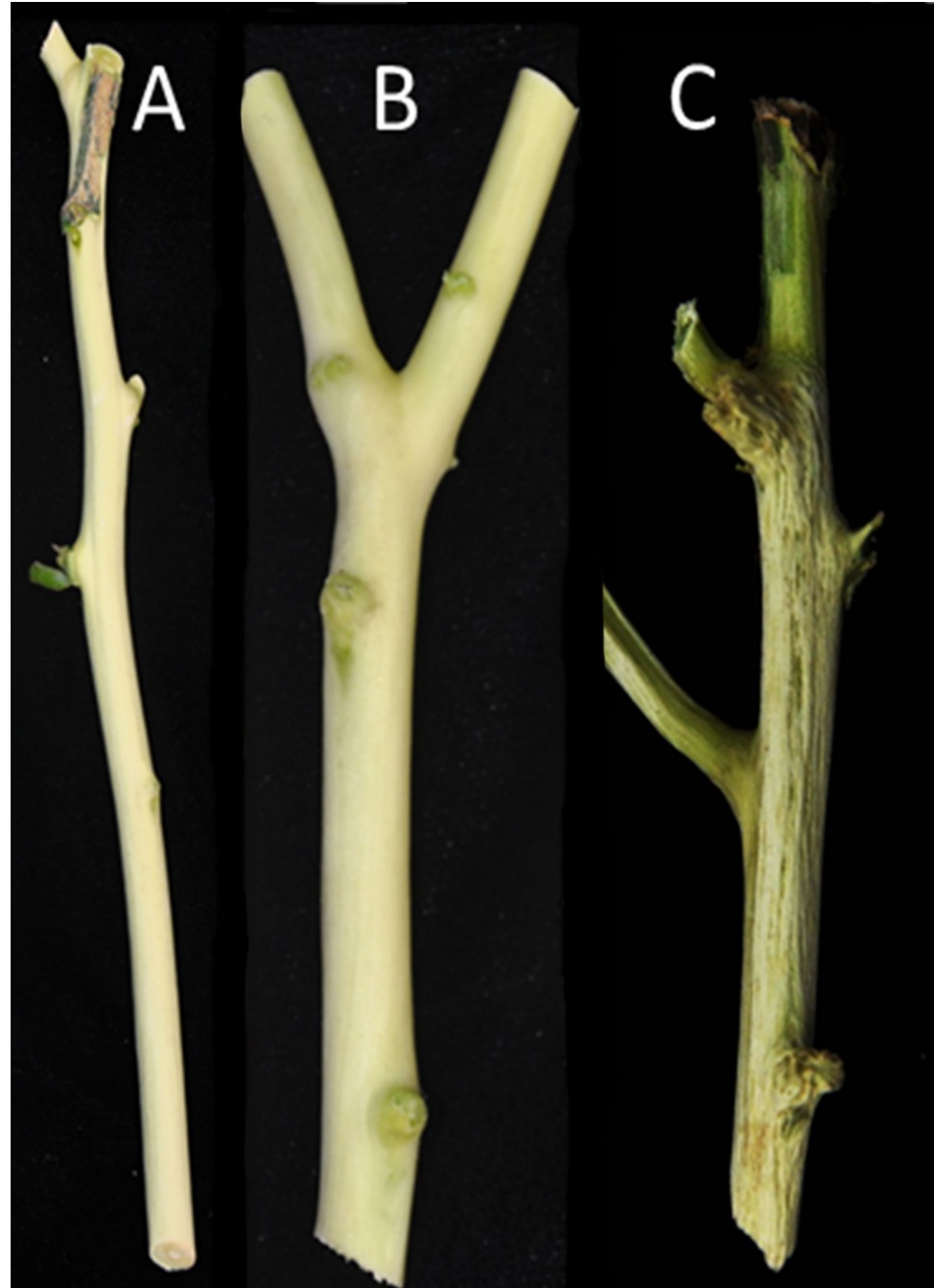

**Fig 1. Comparison of citrus tristeza virus (CTV) strain virulence shown by stem pitting (SP) symptoms in Washington navel.** A) Healthy controls, B) T36-mild with no SP, and C) VT SY568 severe with strong SP and necrosis. Virulence of SP varied seasonally with new growth cycles.

real time quantitative PCR (RT-qPCR) with universal and strain-specific primers and TaqMan probes was performed [29]. CTV titer was monitored seasonally. All plants were maintained in a California Department of Food and Agriculture (CDFA) approved citrus structure under permit and in compliance of State Interior Quarantine §3407 [30].

## Phenotype evaluation

After four years of growth in the screenhouse, each tree was decapitated at ground level and trunk circumference was measured at 30 cm above the bud union. Tree biomass was measured by wet weight of the entire tree. Stem pitting was rated by peeling the trunk above the bud union, as well as peeling branches. The best location to observe stem pitting was in the branches that developed over the past year or two.

## Chemical analyses

One year after infections, all trees utilized in this study had three leaves collected and immediately flash-frozen in liquid nitrogen. Samples were stored at -20°C until further processing. Samples were processed by pulverizing leaf tissues in liquid nitrogen with a mortar and pestle. A total of 0.1 g of pulverized leaf tissue was then weighed and placed into each of two separately labeled 1.5 mL microcentrifuge tubes. One of these tubes had the tissue twice extracted in 0.5 mL of methanol (Thermo-Fisher, Pittsburgh, USA) at 4°C overnight each extraction, with the two extractions combined for a total of 1 mL of methanol extract [31]. Another tube had the tissue twice extracted in 0.5 mL of methyl tert-butyl ether (MTBE) (Millipore-Sigma, St. Louis, USA), with 0.1% (v/v) n-pentadecane (Millipore) added as an internal standard, at 4°C overnight each extraction, for a total of 1 mL of MTBE extract [31].

To analyze phenolic compounds, 50 μL of the methanol extract was injected into a Shimadzu (Columbia, MD, USA) LC-20AD pump-based high-performance liquid chromatography (HPLC) system with a Supelco Ascentis RP-18 (25 cm x 4.6 mm internal diameter x 5 μm particle size), kept at 50°C in a column oven, and read with a Shimadzu SPD-20 photodiode array detector. A binary gradient was used to proceed from 95% solvent A (water with 0.2% (v/v acetic acid) (Millipore) to 100% solvent B (methanol with 0.2% acetic acid) and back over 40 minutes, according to [32]. Peaks were read at 280 nm. Similar conditions were utilized to run select samples for each treatment combination on a Shimadzu HPLC system equipped with a Shimadzu LC-MS2020 mass spectrometer. This provided mass spectra to be obtained for each peak and, along with UV/Visible spectra and retention times, allowed putative peak identifications when compared with similar information in published literature [32–34]. A selection of standards was obtained from Millipore-Sigma to create standard curves to convert compounds to mg/g fresh weight amounts for each major compound class and to definitively identify certain peaks.

To analyze terpenoid compounds, 2 μL of the MTBE extract was injected into a Shimadzu QP2010S gas chromatograph-mass spectrometer (GC-MS) equipped with a SHRXI-5MS column (30 cm x 0.25 mm x 0.25 μm) and using helium as a carrier gas at a constant linear velocity of 30 cm/s. The oven gradient went from 60°C to 200°C over 35 min, then to 250°C over 8 min and 20 s, followed by a hold at 250°C for 6 min and 40 s, for a total runtime of 45 min [34]. Peaks were identified and quantified by commercial standards obtained from Millipore-Sigma.

To analyze amino acids, 100 μL of the methanol extract was used with the Phenomenex (Torrance, CA, USA) EZ-FAAST amino acid kit for gas chromatography with a flame ionization detector (GC-FID) and the provided Phenomenex Zebron AA column. Procedures from the EZ-FAAST kit were followed strictly, and the included internal and external standards were used to identify and quantify derivatized compound peaks.

To analyze sugars/carbohydrates, 50 μL of the methanol extract was injected into a Shimadzu (Columbia, MD, USA) LC-10AD pump based high-performance liquid chromatography system with a Supelcogel (Millipore-Sigma) H column (25 cm x 4.8 mm internal diameter x 9 μm particle size) held in a column heater at 40°C. The solvent was 0.1 M phosphoric acid

(Thermo-Fisher) and was delivered isocratic at 0.1 mL/min for 60 minutes per sample. Results were recorded on Shimadzu RID-10 Refractive Index Detector. Standards curves were made for sucrose, glucose, and fructose running the same conditions as the samples.

### Statistical analyses

Analyses of variance (ANOVA), when normality assumptions were met as assessed by residual plots, were used to determine treatment effects of cultivar, CTV infection status, and the interaction on CTV titers, symptoms, and total levels of each compound class (phenolics, terpenoids, or amino acids) and individual sugar levels (fructose and glucose). Tukey HSD tests were utilized for subsequent pairwise comparisons. Kruskal-Wallis or Mann-Whitney U tests were used when normality assumptions were not met. For individual compounds, MANOVAs were performed using cultivar, CTV infection status, and the interaction as variables, followed by ANOVAs on individual compounds then separations by Tukey HSD tests. All of these statistics were performed in SPSS version 24.0 (IBM, Armonk, NY, USA).

Furthermore, metabolite profiling was conducted to reveal differential accumulation of select phenolics, terpenoids, amino acids, and sugars to see if these compounds could be used to discriminate citrus species, as well as between healthy and CTV-infected plants. Thus, stepwise linear canonical discriminant analysis (CDA) was performed using the statistical program JMP Pro version 15.2.0 (SAS Institute Inc., NC, USA). CDA utilized metabolite data from each citrus species of mock-inoculated controls (healthy) and CTV-inoculated (both mild and severe) trees with either phenolics, terpenoids, or amino acids as dependent variables. Only the top ten variables of each class were used to prevent over-fitting of the models. The models were self-tested with the same data to determine their accuracy in identification of citrus species and infection status.

## Results

### Cultivar and CTV infection status effects on observed titer, stem pitting, and plant growth

For the purpose of simplicity, CTV strains were organized into two classes: mild and severe (Table 1). ANOVA observed that titers of the two CTV classes had lower Ct values, which indicates greater titers, than healthy controls ($F_{2, 162}$ = 755.904; $P < 0.001$), and severe CTV strains had even lower Ct values than mild strains. Citrus species also significantly affected titers, with Oro Blanco supporting lower CTV titers while Navel and Valencia sweet oranges possessed high titers ($F_{5, 162}$ = 3.532; $P$ = 0.005) (Fig 2). There was not a significant CTV class by citrus type interaction ($F_{10, 162}$ = 1.322; $P$ = 0.223). Regardless, effects of CTV classes on titers within

**Table 1. Mild and severe CTV strains utilized in this study.**

| CTV Characterization | CTV Isolate Name | CTV Strain Type |
|---|---|---|
| Mild | CCTEA 65 | S1-mild |
| | P09.1b | T36-mild |
| | Mur-AT4 | T30-mild |
| | CCTEA 115 | RB-mild |
| Severe | CCTEA 441 | VT-severe |
| | RH | VT-severe |
| | Svl | VT-severe |
| | SY568 | VT-severe |
| | P108-AT39 | VT-severe |

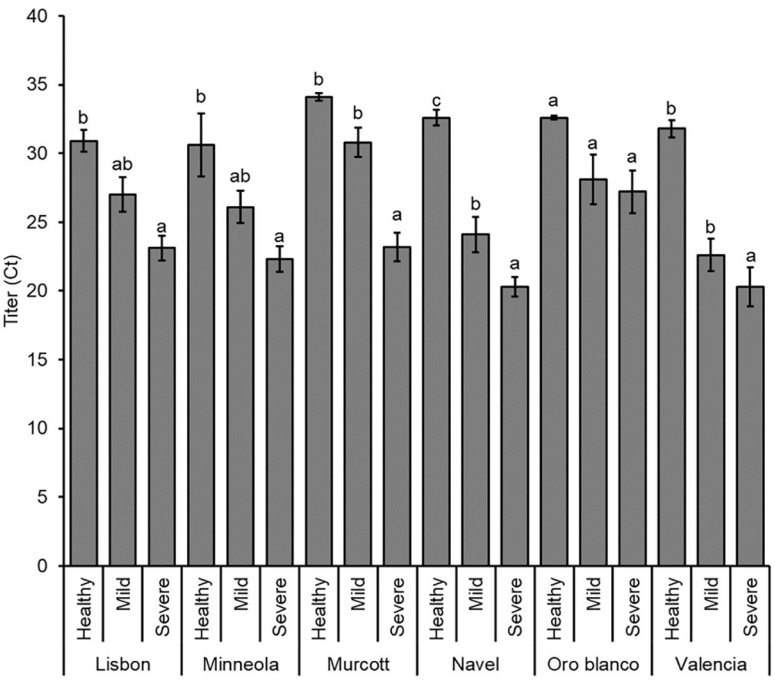

**Fig 2. CTV titers.** CTV titers (± SE) (as Ct values from real-time PCR, with lower values representing greater titers) in healthy, CTV-mild strain infected, and CTV-severe strain infected citrus trees. Different letters represent significantly different means by Tukey HSD tests for each citrus species individually.

each citrus class were performed, with overall differences generally being severe strains having significantly ($P < 0.05$) higher titer than mild strains classes, except for Oro Blanco (Fig 2).

Non-parametric Kruskal-Wallis tests revealed significant differences among cultivars in stem pitting development ($X^2_{5, 113} = 20.300$; $P = 0.001$). Follow-up Mann-Whitney U tests ($P < 0.05$) revealed more stem pitting with severe CTV in all other citrus species than Lisbon, and more stem pitting in Oro Blanco than Murcott (Fig 3A). Kruskal-Wallis tests and Mann-Whitney U tests also observed more stem pitting in severe CTV strains than mild strains or controls ($X^2_{2, 116} = 31.698$; $P < 0.001$) (Fig 3A).

Tree weight differed due to citrus cultivar ($F_{5, 101} = 35.545$; $P < 0.001$), with Lisbon trees weighing more than all other species trees, Oro Blanco weighing more than all other species except Lisbon, and Murcott trees weighing more than Navel trees (Fig 3B). Tree weight also differed due to CTV class ($F_{2, 101} = 4.949$; $P = 0.009$), with greater weights in trees infected with mild CTV strains than healthy controls (Fig 3B). There was no significant citrus cultivar by CTV class interaction effect on tree weight ($F_{10, 101} = 1.643$; $P = 0.105$).

Tree trunk diameter at 30 cm from the soil line did not differ due to CTV class ($F_{2, 101} = 2.346$; $P = 0.101$), but did vary due to citrus species ($F_{5, 101} = 21.094$; $P < 0.001$), with Oro Blanco having greater diameters than the other tree species except Lisbon, and Lisbon and Valencia having greater diameters than Minneola, Murcott and Navel (Fig 3C). The citrus type and CTV type interaction was significant ($F_{10, 101} = 3.253$; $P = 0.001$), but no discernable pattern was observed behind it.

## Cultivar and CTV infection status effects on phenolic levels

Total phenolic levels differed among citrus cultivars ($F_{5, 162} = 68.650$; $P < 0.001$), with Oro Blanco possessing more phenolic levels than all other citrus, and Lisbon and Murcott with

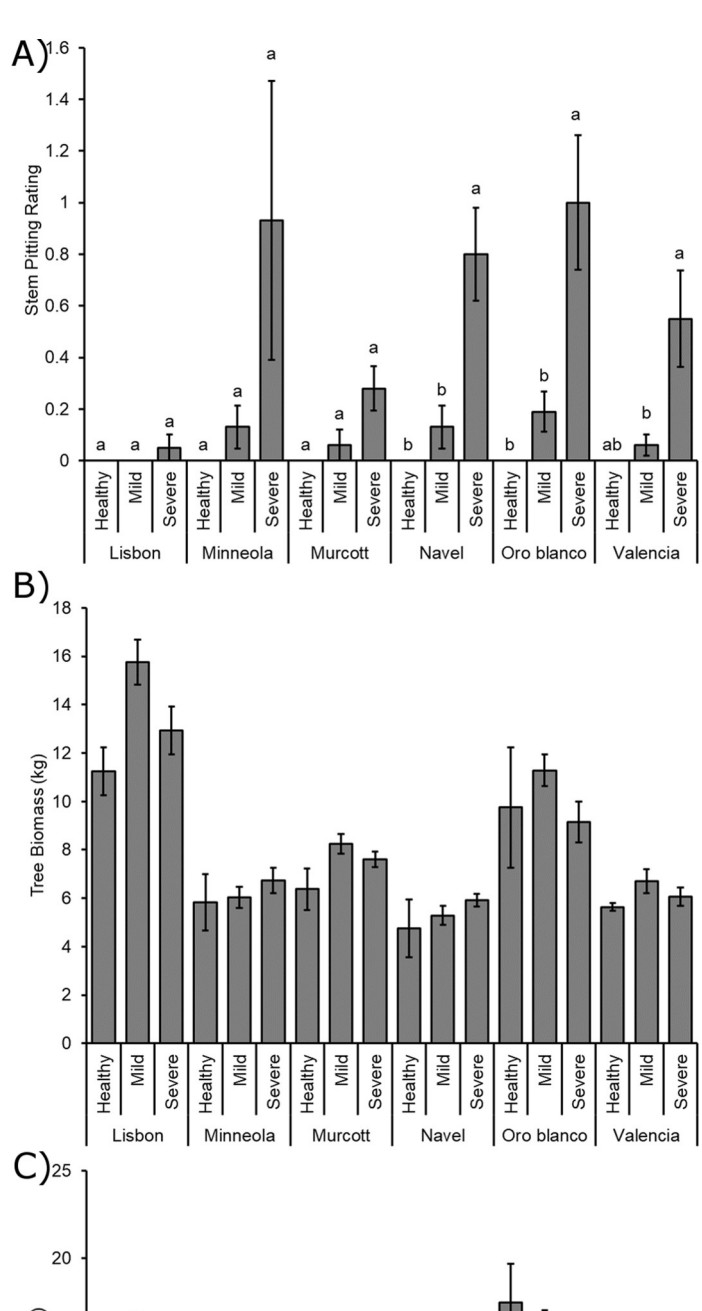

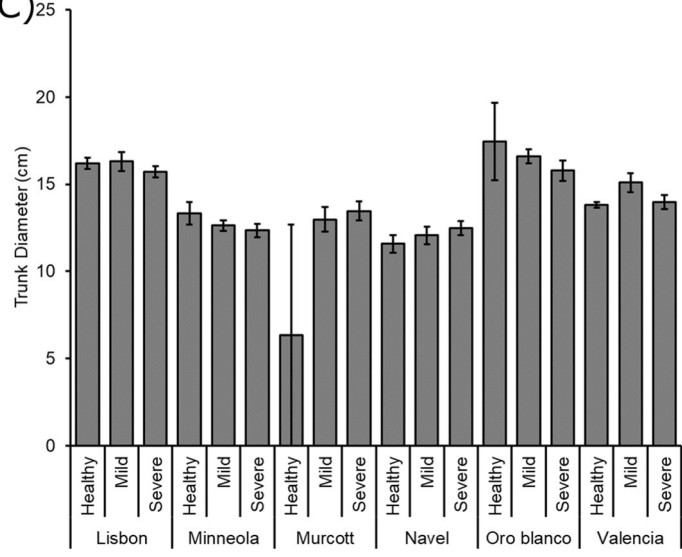

**Fig 3. Disease and tree growth measurements.** Disease and growth measurements of healthy and CTV infected trees. A) Stem pitting rating (0–5) (± SE) for each citrus cultivar that was healthy, CTV-mild strain infected, or CTV-severe strain infected. Different letters represent significantly different means by Mann-Whitney U tests for each citrus species individually. B) Mean (± SE) biomass weight for healthy and infected trees. C) Mean (± SE) trunk diameter at 30 cm from soil level for healthy and infected trees.

more phenolic levels than Minneola, Navel, and Valencia (Fig 4). Total phenolic levels were not different due to infection status ($F_{2, 162} = 2.296$; $P = 0.104$). There was not a significant citrus cultivar by CTV status interaction ($F_{10, 162} = 1.028$; $P = 0.422$).

MANOVA of individual phenolic compounds revealed significant effects of cultivar (Pillai's trace = 4.470; $F = 35.549$; $P < 0.001$), CTV infection status (Pillai's trace = 0.931; $F = 3.596$; $P < 0.001$), and the cultivar by CTV status interaction (Pillai's trace = 3.099; $F = 1.964$; $P < 0.001$). Follow-up ANOVA tests revealed significant effects of cultivar on every phenolic compound (Fig 5 and Table A in S1 Appendix). Lisbon had significantly greater levels of 14 phenolics (apigenin 6-C-glucosyl-7-O-(6-malyl-glucoside), didymin, eriocitrin, hesperidin, isorhoifolin-4-glucoside, lucenin-2, lucenin-2 4-methyl ether, luteolin-7-rutinoside, naringin, narirutin, stellarin-2, vicenin-2, unknown flavanone 1, and unknown flavone 2) than all other cultivars, and Murcott had significantly greater levels of 10 phenolics (heptamethoxyflavone, hexamethoxyflavone, hexamethyl-O-quercetagetin, isosinensetin, natsudaidain, nobiletin, sinensetin, tangeretin, tetramethyl-O-scutellarein, unknown flavonol methyl ester, and unknown polymethoxylated flavone) than most other cultivars (Table A in S1 Appendix).

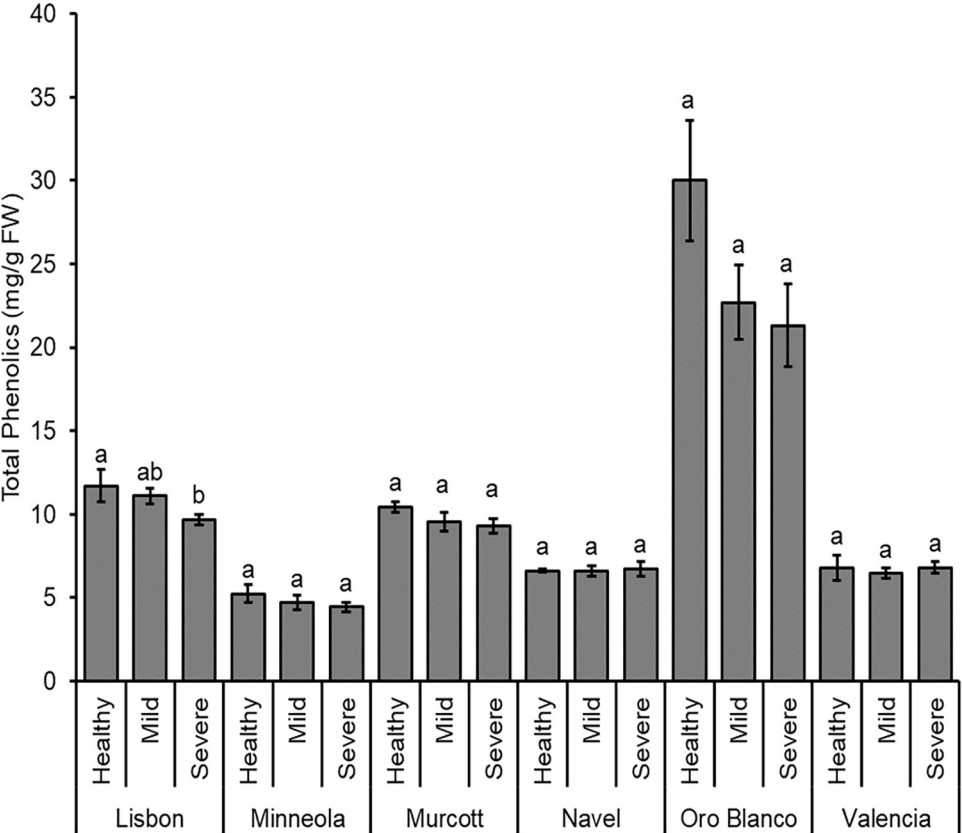

**Fig 4. Summed total of all phenolic compounds.** Mean of total phenolic compound levels (± SE) for each citrus cultivar that was health, CTV-mild strain infected, or CTV-severe strain infected. Different letters represent significantly different means by Tukey HSD tests for each citrus species individually.

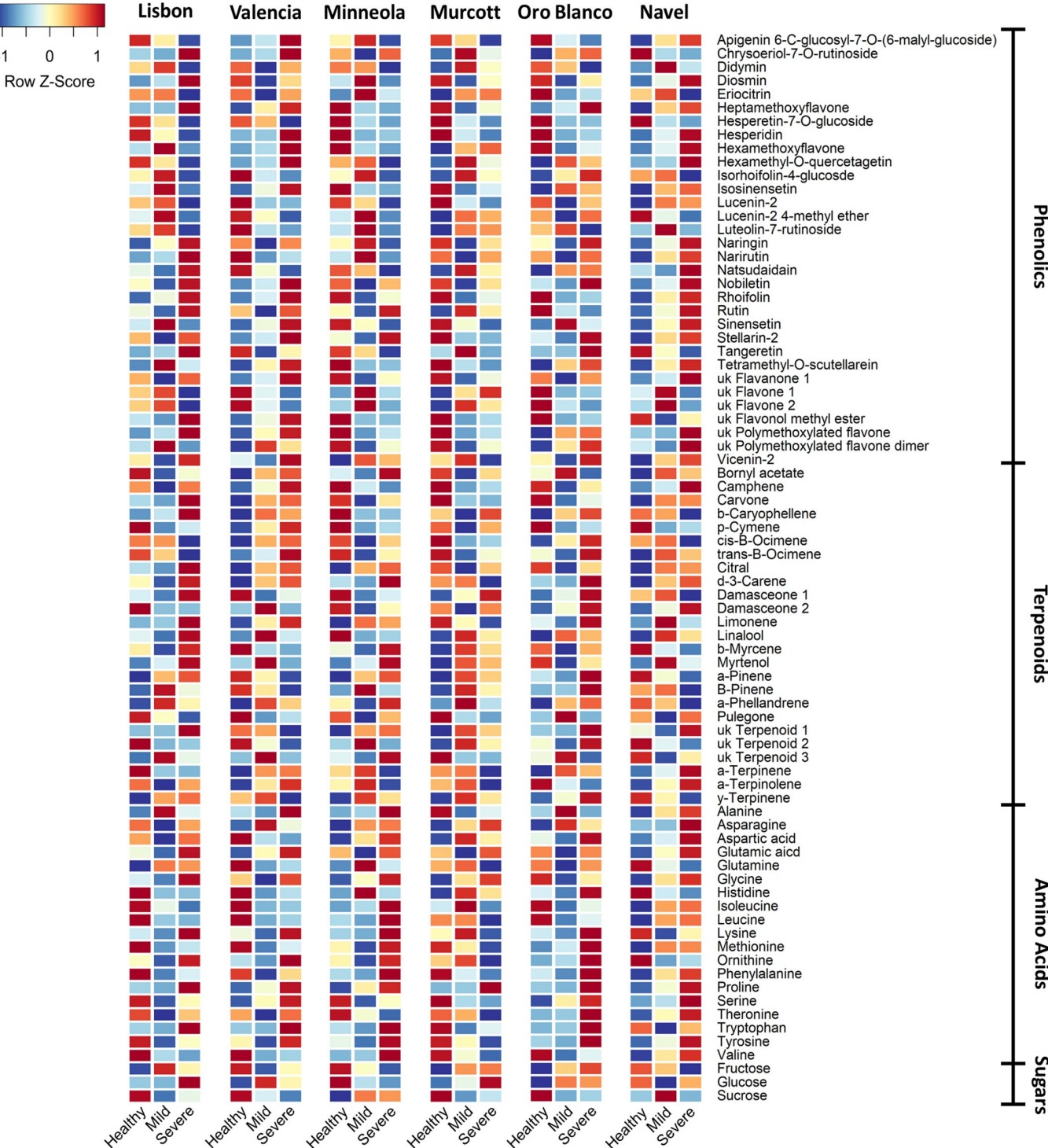

**Fig 5. Heatmap of individual compound levels across treatments.** Heatmap showing different patterns of individual compound levels for A) phenolics, B) terpenoids, or C) amino acids among cultivars and trees with different infection status. For supporting statistics refer to the ANOVA tables provided in the S1 Appendix.

Follow-up ANOVA tests also revealed that CTV status significantly affected levels of phenolic compounds as well (Table A in S1 Appendix). Plants with severe CTV strain infections had greater levels of diosmin, stellarin-2, unknown flavone 1, and unknown polymethoxylated flavone than mild CTV strain infected plants. Mild CTV infected plants had greater levels of heptamethoxyflavone, isorhoifolin-4-glucoside, lucenin-2 4-methyl ether, luteolin-7-rutinoside, nobiletin, and unknown flavone 2 than plants infected with severe CTV strains. Healthy plants had greater levels of diosmin, rhoifolin, and unknown flavonol methyl ester than mild CTV strain infected plants. Lastly, healthy plants had greater levels of hesperetin-7-O-glucoside and unknown flavone 2 than plants infected with severe CTV strains. Follow-up ANOVAs for the cultivar and infection status interactions were significant for diosmin, heptamethoxyflavone, isosinensetin, luteolin-7-rutinoside, natsudaidain, nobiletin, rhoifolin, tangeretin, unknown flavone 1, unknown flavone 2, unknown flavonol methyl ester, unknown polymethoxylated flavone, and unknown polymethoxylated flavone dimer (Table A in S1 Appendix).

### Cultivar and CTV infection status effects on terpenoid levels

Total terpenoid levels were significantly different among citrus types ($F_{5, 162} = 72.928$; $P < 0.001$), but did not differ between CTV infection treatments ($F_{2, 162} = 0.567$; $P = 0.568$). There was no significant interaction between cultivar and CTV infection treatment ($F_{10, 162} = 1.009$; $P = 0.438$). Lisbon had greater terpenoid levels than all other citrus, Murcott had greater amounts than all other citrus except Lisbon, and Navel had greater amounts than Minneola, Oro Blanco, and Valencia (Fig 6).

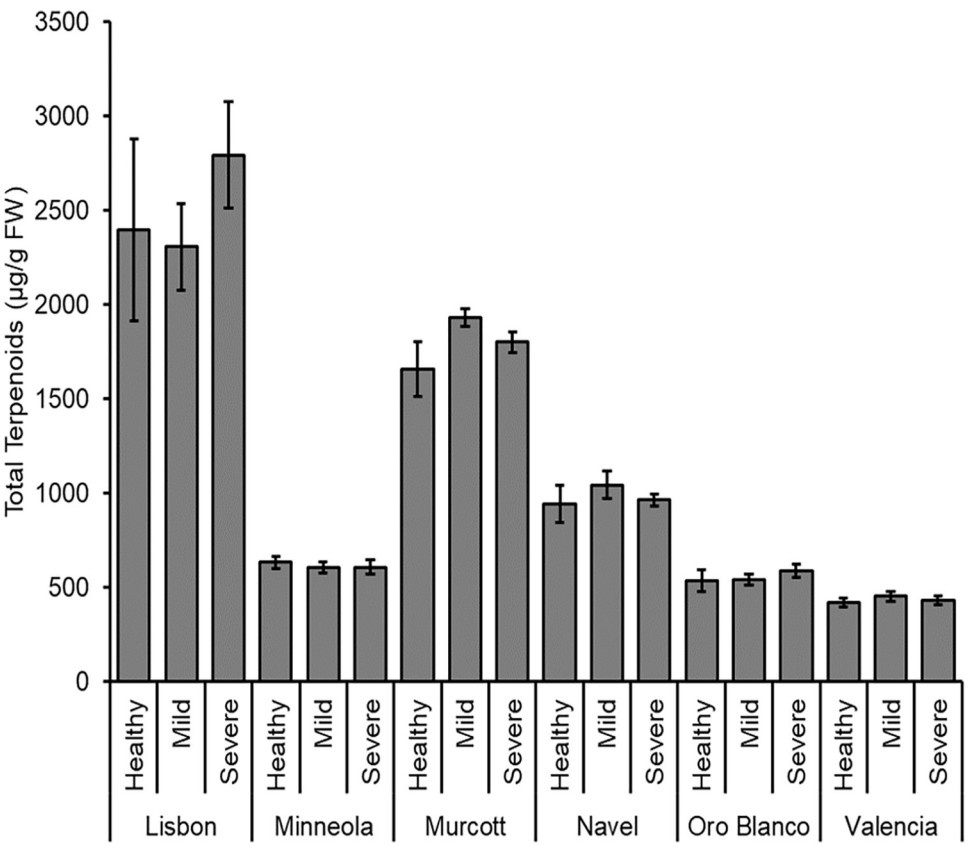

**Fig 6. Summed total of all terpenoid compounds.** Mean of total terpenoid levels (± SE) for each citrus cultivar that was healthy, CTV-mild strain infected, or CTV-severe strain infected.

As with phenolics, MANOVA of individual terpenoid compounds revealed significant effects of cultivar (Pillai's trace = 4.175; $F$ = 28.749; $P$ < 0.001), CTV infection status (Pillai's trace = 0.671; $F$ = 2.807; $P$ < 0.001), and the cultivar by CTV status interaction (Pillai's trace = 2.370; $F$ = 1.826; $P$ < 0.001).

Follow-up ANOVA tests revealed significant effects of cultivar on every terpenoid except α-phellandrene (Fig 5 and Table B in S1 Appendix). Murcott had greater levels of 12 individual terpenoids (α-pinene, α-terpinene, α-terpinolene, cis-β-ocimene, trans-β-ocimene, β-pinene, γ-terpinene, camphene, damasceone isomer 2, linalool, para-cymene, and an unknown/unidentified terpenoid) than most of the other cultivars, and Lisbon had greater levels of 11 individual terpenoids (β-caryophyllene, β-myrcene, trans-β-ocimene, β-pinene, bornyl acetate, camphene, carvone, citral, a damasceone isomer 1, limonene, and myrtenol) than the other cultivars except Murcott. For the remaining compounds, results were variable (Table B in S1 Appendix).

ANOVA tests revealed that CTV infection status significantly affected only α-terpinene (healthy and mild CTV infected plants had greater levels than plants infected with severe CTV strains), β-pinene (mild CTV strain infected plants had greater levels than severe CTV infected strains and controls), linalool (mild CTV infected plants had greater levels than healthy plants), and para-cymene (severe CTV strain infected plants had greater levels than mild CTV infected plants) (Table B in S1 Appendix). Follow-up ANOVAs for the cultivar and infection status interactions were significant for α-terpinene, α-terpinolene, linalool, para-cymene, and pulegone (Table B in S1 Appendix).

## Cultivar and CTV infection status effects on plant amino acid and sugar levels

Total amino acid levels differed among the different citrus cultivars ($F_{5,\ 161}$ = 28.611; $P$ < 0.001), with Murcott possessing more amino acids than all other citrus (Fig 7). Navel had more amino acids than Lisbon, Oro Blanco, and Valencia. Minneola had more than Lisbon and Oro Blanco. Oro Blanco had the least amino acids of any citrus cultivar tested.

CTV status also affected total amino acids levels across all cultivars, with citrus infected with severe CTV strains possessing greater levels of total amino acids than citrus infected with mild CTV strains or healthy trees ($F_{2,\ 161}$ = 17.708; $P$ < 0.001). There was no significant citrus cultivar by CTV infection status interaction ($F_{10,\ 161}$ = 1.729; $P$ = 0.078).

MANOVA on individual amino acids revealed significant effects of cultivar (Pillai's trace = 3.823; $F$ = 25.116; $P$ < 0.001), CTV infection status (Pillai's trace = 0.722; $F$ = 4.285; $P$ < 0.001), and the cultivar by CTV infection status interaction (Pillai's trace = 2.159; $F$ = 2.202; $P$ < 0.001). Follow-up ANOVA tests revealed significant effects of cultivar on every amino acid (Fig 5 and Table C in S1 Appendix). For almost every amino acid (with the exception of asparagine), Murcott had significantly greater levels than at least three of the other cultivars. Oro Blanco had significantly less amino acid amounts of every individual amino acid except asparagine. For asparagine, Minneola had greater levels than all other cultivars except Navel, and Navel had greater levels than Lisbon and Valencia.

ANOVA tests revealed that CTV infection type significantly affected eight individual amino acids (Table C in S1 Appendix). Of these, Tukey tests did not observe differences between strains for glutamine or methionine. However, leaves infected by severe CTV strains had greater levels of glutamic acid, glycine, phenylalanine, proline, tryptophan, and tyrosine than mild strains. Glutamic acid and proline levels also were greater in CTV severe strain-infected leaves than non-infected controls. Follow-up ANOVAs for the cultivar and infection

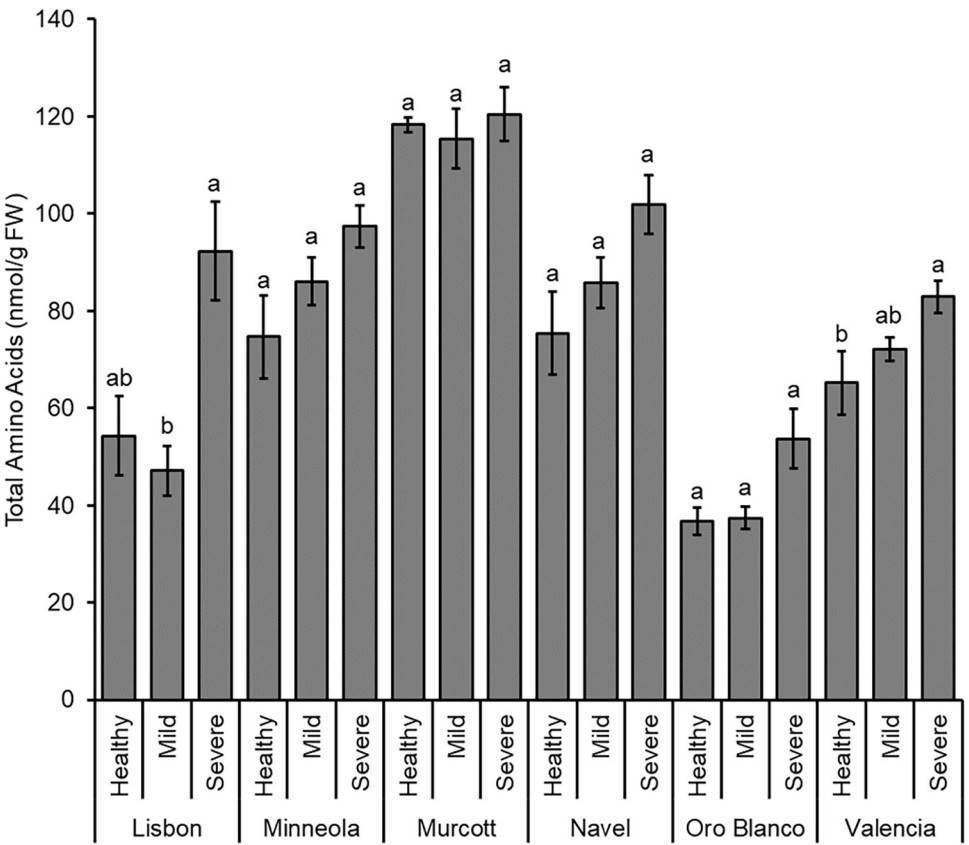

**Fig 7. Summed total of all amino acid compounds.** Mean of total amino acid levels (± SE) for each citrus cultivar that was health, CTV-mild strain infected, or CTV-severe strain infected. Different letters represent significantly different means by Tukey HSD tests for each citrus species individually.

status interactions were significant for glutamine, isoleucine, leucine, lysine, phenylalanine, proline, threonine, tyrosine, and valine (Table C in S1 Appendix).

Fructose levels were significantly affected by cultivar ($F_{5, 162}$ = 7.020; $P < 0.001$), but not CTV infection status ($F_{2, 162}$ = 2.121; $P = 0.123$), and had a significant cultivar and CTV infection status interaction ($F_{10, 162}$ = 2.881; $P = 0.002$). Greater fructose levels were present in Minneola (107 ± 6 mg/g) and Valencia (100 ± 5 mg/g) trees than Navel (80 ± 4 mg/g) and Oro Blanco (82 ± 4 mg/g). Lisbon (90 ± 5 mg/g) and Murcott (94 ± 4 mg/g) trees were in between. Glucose levels were observed to be different among cultivars ($F_{5, 162}$ = 6.007; $P < 0.001$) but not due to CTV infection status ($F_{2, 162}$ = 0.837; $P = 0.435$). There was a significant cultivar by infection status interaction ($F_{15, 162}$ = 4.014; $P < 0.001$). Greater glucose levels were present in Lisbon trees (273 ± 15 mg/g; mean ± SE) than Murcott (206 ± 8 mg/g) and Oro Blanco (218 ± 15 mg/g). Levels in Minneola (256 ± 17 mg/g), Navel (236 ± 14 mg/g) and Valencia (227 ± 11 mg/g) were in between.

## Canonical discriminant analysis (CDA) of CTV infection status in different citrus cultivars based on metabolite profiling

Metabolite profiling of citrus infected with different CTV strains revealed differential accumulation of select phenolics, terpenoids, amino acids, and sugars among citrus of differing CTV infection status. Thus, this robust metabolomic data set was used to see if phenolics,

terpenoids, and amino acids could be used to discriminate between different citrus cultivars and different CTV status within a cultivar. Furthermore, the most important metabolites within these classes for discrimination was identified. To accomplish this, stepwise linear canonical discriminant analysis (CDA) was performed of different citrus cultivars and healthy and mild- and severe-CTV infected citrus, using select phenolics, terpenoids, and amino acids as dependent variables. Only the ten most variable metabolites of each class were used in CDAs to prevent over-fitting of the models. These metabolites are listed in Table 2 and

**Table 2. Ten most variable phenolics, terpenoids, and amino acids identified in CDA.**

| Citrus Cultivar | Phenolics | Terpenoids | Amino Acids |
|---|---|---|---|
| Lisbon | uk Flavone 1 | uk terpenoid 1 | Alanine |
| | Naringin | α-Pinene | Glycine |
| | Eriocitrin | β-Myrcene | Serine |
| | uk Flavanone 1 | δ-3-Carene | Proline |
| | Hesperidin | para-Cymene | Aspartic acid |
| | uk Polymethoxylated flavone dimer | cis-β-Ocimene | Methionine |
| | uk Polymethoxylated flavone | trans-β-Ocimene | Glutamic acid |
| | Isosinensetin | α-Terpinene | Ornithine |
| | Nobiletin | Pulegone | Tyrosine |
| | Tangeretin | Damasceone 2 | Tryptophan |
| Minneola | uk Flavone 2 | α-pinene | Glycine |
| | Lucenin-2 | Camphene | Isoleucine |
| | Vicenin-2 | δ-3-Carene | Theronine |
| | uk Flavanone 1 | para-Cymene | Serine |
| | Lucenin-2 4-methyl ether | Linalool | Proline |
| | uk Polymethoxylated flavone dimer | Carvone | Aspartic acid |
| | uk Flavonol methyl ester | β-Caryophellene | Glutamic acid |
| | Sinensetin | uk terpenoid 3 | Glutamine |
| | Nobiletin | γ-Terpinene | Ornithine |
| | Natsudaidain | Damasceone 2 | Tyrosine |
| Murcott | uk Flavone 1 | Camphene | Valine |
| | Luteolin-7-rutinoside | β-Myrcene | Leucine |
| | Lucenin-2 | para-Cymene | Serine |
| | Vicenin-2 | Limonene | Proline |
| | Eriocitrin | Linalool | Aspartic acid |
| | Stellarin-2 | α-Terpinene | Methionine |
| | Diosmin | Pulegone | Glutamic acid |
| | uk Polymethoxylated flavone dimer | Bornyl acetate | Lysine |
| | Nobiletin | Damasceone 1 | Tyrosine |
| | Natsudaidain | Damasceone 2 | Tryptophan |
| Navel | uk Flavone 2 | β-Myrcene | Alanine |
| | Apigenin 6-C-glucosyl-7-O-(6-malyl-glucoside) | δ-3-Carene | Glycine |
| | Naringin | Limonene | Theronine |
| | Hesperetin-7-O-glucoside | cis-β-Ocimene | Serine |
| | Hesperidin | trans-β-Ocimene | Proline |
| | Diosmin | α-Terpinolene | Asparagine |
| | Chrysoeriol-7-O-rutinoside | Linalool | Glutamic acid |
| | uk Flavonol methyl ester | Citral | Glutamine |
| | Hexamethyl-O-quercetagetin | Myrtenol | Histidine |

*(Continued)*

**Table 2.** (Continued)

| Citrus Cultivar | Phenolics | Terpenoids | Amino Acids |
|---|---|---|---|
| | Nobiletin | Damasceone 2 | Tryptophan |
| Oro Blanco | Luteolin-7-rutinoside | Camphene | Leucine |
| | Isorhoifolin-4-glucosde | β-Pinene A | Isoleucine |
| | Lucenin-2 | β-Myrcene | Theronine |
| | Rhoifolin | α-Phellandrene | Serine |
| | Hesperidin | para-Cymene | Proline |
| | Rutin | Citral | Methionine |
| | Lucenin-2 4-methyl ether | β-Caryophellene | Glutamic acid |
| | Diosmin | α-Terpinene | Phenylalanine |
| | uk Polymethoxylated flavone dimer | Pulegone | Ornithine |
| | uk Favonol methyl ester | Bornyl acetate | Tyrosine |
| Valencia | Luteolin-7-rutinoside | α-Pinene A | Alanine |
| | Isorhoifolin-4-glucosde | β-Pinene | Valine |
| | Vicenin-2 | β-Myrcene | Isoleucine |
| | Didymin | α-Phellandrene | Serine |
| | Hesperetin-7-O-glucoside | δ-3-Carene | Proline |
| | Stellarin-2 | cis-β-Ocimene | Asparagine |
| | Lucenin-2 4-methyl ether | Carvone | Aspartic acid |
| | Sinensetin | β-Caryophellene | Glutamic acid |
| | Tetramethyl-O-scutellarein | Pulegone | Phenylalanine |
| | Heptamethoxyflavone | Damasceone 2 | Tyrosine |

The 10 most variable phenolics, terpenoids, and amino acids identified in CDA of mock-inoculated (healthy) and CTV-infected plants within each cultivar.

Table A in S1 Appendix. CDAs of different cultivars with phenolics, terpenoids, and amino acids were able to predict the correct cultivar of each sample with at least 98% accuracy for each metabolite class (Table B in S1 Appendix). This highlights the significant differences of these various metabolites across the difference citrus cultivars used. Due to the significant differences, CDA of CTV status was done separately for each cultivar. Consequently, the optimal phenolics, terpenoids, and amino acids for CDA differed between each citrus cultivar (Table 2).

Canonical plots generated from CDA utilizing phenolics revealed that healthy citrus tended to cluster further from CTV-infected citrus, with citrus infected with mild and severe strains of CTV clustering more closely together (Fig 8). This was true in most cultivars, except in Lisbon and Navel. Though mild-CTV citrus tended to cluster more closely with severe-CTV citrus, they still effectively clustered separately. Samples belonging to a particular CTV status clustered correctly, with at least 97% overall accuracy across cultivars (Table 3).

CDA of CTV status with terpenoids produced similar results compared to phenolics. Canonical plots generated from CDA using terpenoids show that all treatment types clustered separately from one another, with mild and severe CTV-infected citrus clustering more closely together (Fig 9), just as with phenolics. This was true for most cultivars, except Navel and Oro Blanco. Overall, samples were assigned to the correct treatment within each cultivar with high accuracy (Table 4). Most cultivars had their samples correctly assigned with 97% accuracy, while Minneola and Navel displayed 90% and 87% correct assignments of samples, respectively.

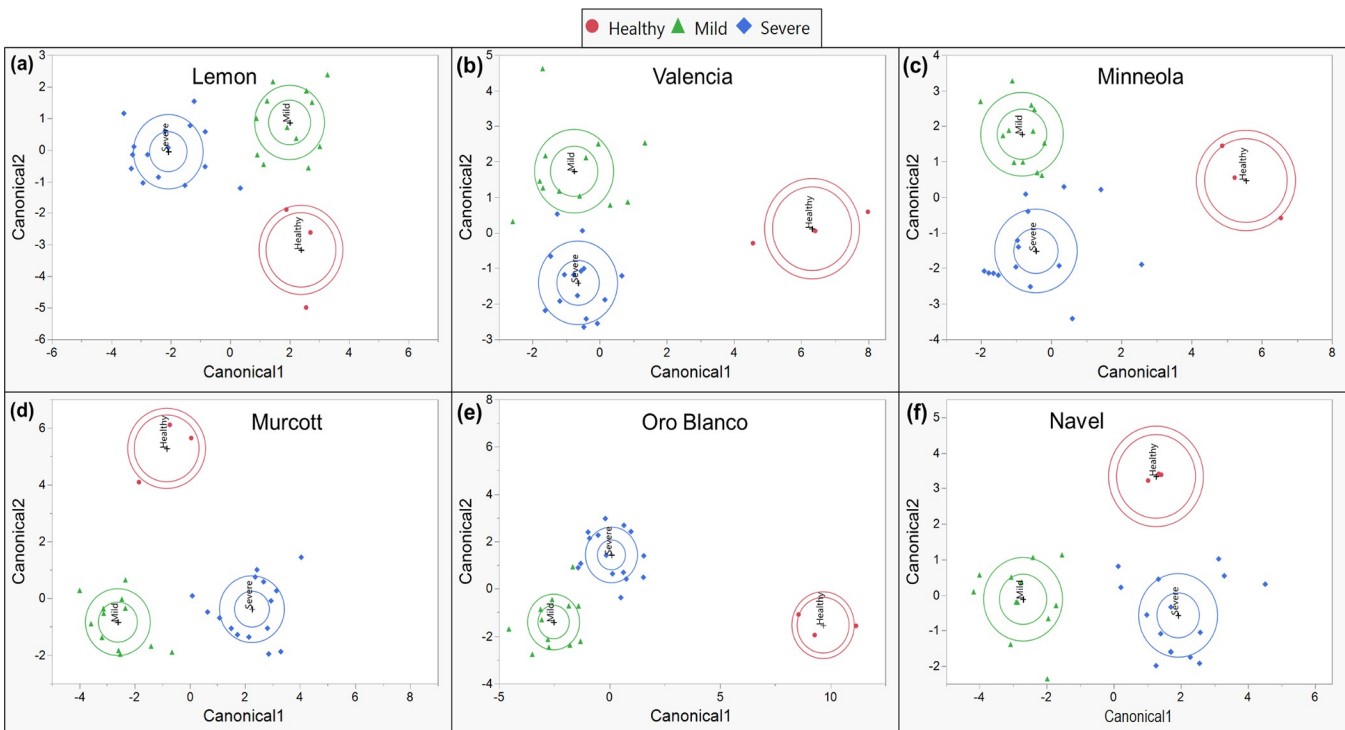

**Fig 8. Canonical plots using 10 most variable individual phenolic compounds.** Canonical plots of different plants displaying mild (green-triangle), severe (blue-diamond), or no (red-circle) symptoms after CTV- or mock-inoculation, within each citrus cultivar. Canonicals were generated by stepwise discriminant analysis utilizing the 10 most variable phenolics between differentially infected plants of each cultivar. Outer ellipses are estimated 50% of the samples within a cultivar, and inner ellipses show the 95% confidence region of cultivar means.

**Table 3. CDA utilizing phenolics.**

| Cultivar | CTV Infection Status | Predicted Count | | | Accuracy | |
|---|---|---|---|---|---|---|
| | | Healthy | Mild | Severe | Match % | Overall % |
| Lisbon | Healthy | 3 | 0 | 0 | 100% | 97% |
| | Mild | 0 | 12 | 0 | 100% | |
| | Severe | 0 | 1 | 14 | 93% | |
| Minneola | Healthy | 3 | 0 | 0 | 100% | 97% |
| | Mild | 0 | 12 | 0 | 100% | |
| | Severe | 0 | 1 | 14 | 93% | |
| Murcott | Healthy | 3 | 0 | 0 | 100% | 100% |
| | Mild | 0 | 12 | 0 | 100% | |
| | Severe | 0 | 0 | 15 | 100% | |
| Navel | Healthy | 3 | 0 | 0 | 100% | 100% |
| | Mild | 0 | 12 | 0 | 100% | |
| | Severe | 0 | 0 | 15 | 100% | |
| Oro Blanco | Healthy | 3 | 0 | 0 | 100% | 97% |
| | Mild | 0 | 11 | 1 | 92% | |
| | Severe | 0 | 0 | 15 | 100% | |
| Valencia | Healthy | 3 | 0 | 0 | 100% | 97% |
| | Mild | 0 | 12 | 0 | 100% | |
| | Severe | 0 | 1 | 14 | 93% | |

CDA utilizing phenolics between healthy, mild CTV infected, and severe CTV infected trees within each citrus cultivar. Different citrus cultivars were either healthy (mock-inoculated) or infected with mild or severe CTV strains.

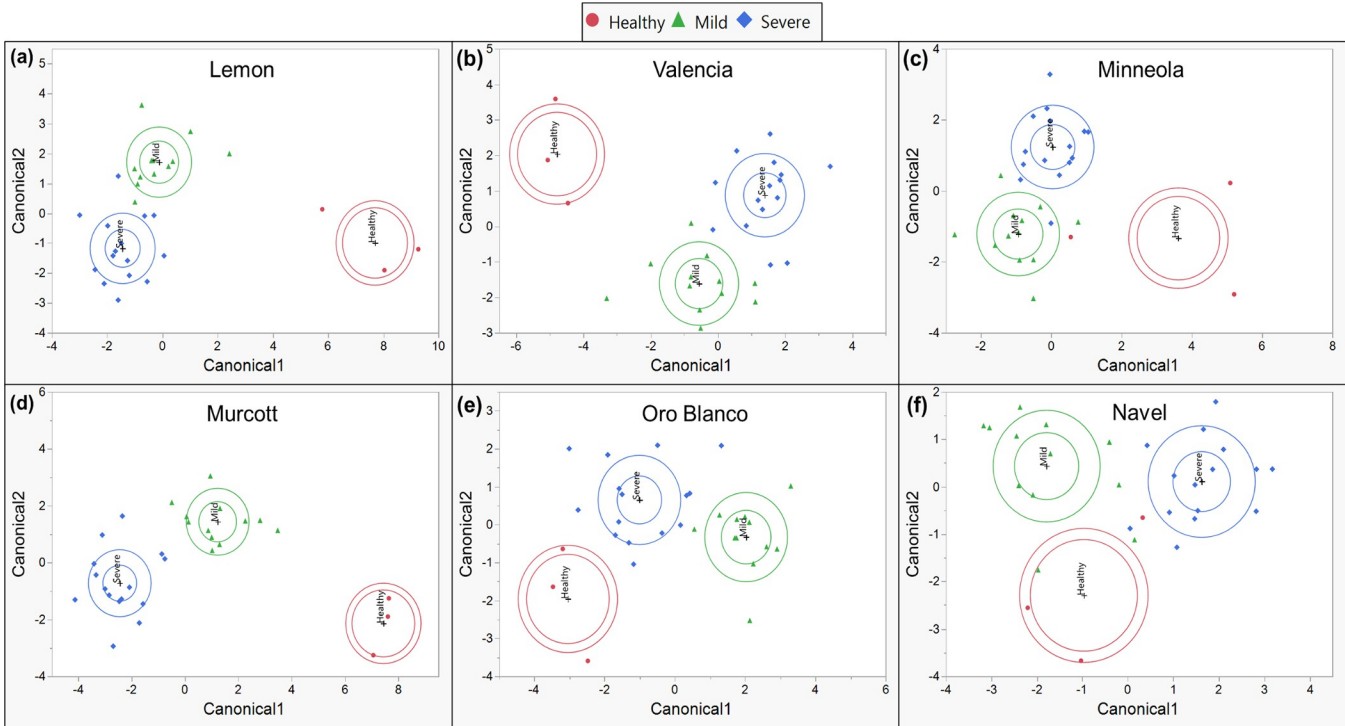

**Fig 9. Canonical plots using 10 most variable individual terpenoid compounds.** Canonical plots of different plants displaying mild (green-triangle), severe (blue-diamond), or no (red-circle) symptoms after CTV- or mock-inoculation, within each citrus cultivar. Canonicals were generated by stepwise discriminant analysis utilizing the 10 most variable terpenoids between differentially infected plants of each cultivar. Outer ellipses are estimated 50% of the samples within a cultivar, and inner ellipses show the 95% confidence region of cultivar means.

CDA utilizing amino acids showed that mild- and severe-CTV infected citrus tended to cluster more closely together than with healthy plants, except for Navel oranges (Fig 10). Despite this, all treatments clustered sufficiently far from one another in all cultivars, with samples of half the cultivars clustering correctly with 97% accuracy (Table 5). Samples from the other half of the cultivars clustered with 90–93% accuracy. Overall, these data suggest that all three groups of metabolites can be used to effectively discriminate CTV health status in citrus, though phenolics and terpenoids were more effective than amino acids. These results highlight the changes these groups of metabolites in citrus collectively undergo infection by CTV.

## Discussion

Assessment of CTV infections revealed that strains categorized as severe did significantly reach greater viral titers than mild strains, albeit some variation existed depending on cultivar. Likewise, severe CTV strains significantly had greater stem pitting than mild strains.

Metabolomic profiling of the major chemical groups (phenolics, terpenoids, amino acids and sugars) revealed substantial variability among different cultivars that was much more significant than differences between CTV treatments, as evidenced by significant interactions in many of the ANOVAs and MANOVAs. Indeed, phenolic compounds [35] and terpenoids [24] were already known to differ greatly between the different citrus cultivars used in this study (grapefruit, lemon, mandarin, and sweet oranges). Because of the disparity in variability of compounds, it was necessary to evaluate metabolite differences within specific cultivars.

**Table 4. CDA utilizing terpenoids.**

| Cultivar | CTV Infection Status | Predicted Count | | | Accuracy | |
|---|---|---|---|---|---|---|
| | | Healthy | Mild | Severe | Match % | Overall % |
| Lisbon | Healthy | 3 | 0 | 0 | 100% | 97% |
| | Mild | 0 | 12 | 0 | 100% | |
| | Severe | 0 | 1 | 14 | 93% | |
| Minneola | Healthy | 2 | 1 | 0 | 67% | 90% |
| | Mild | 0 | 11 | 1 | 91% | |
| | Severe | 0 | 1 | 14 | 93% | |
| Murcott | Healthy | 3 | 0 | 0 | 100% | 100% |
| | Mild | 0 | 12 | 0 | 100% | |
| | Severe | 0 | 0 | 15 | 100% | |
| Navel | Healthy | 2 | 0 | 1 | 67% | 87% |
| | Mild | 2 | 10 | 0 | 83% | |
| | Severe | 1 | 0 | 14 | 93% | |
| Oro Blanco | Healthy | 3 | 0 | 0 | 100% | 97% |
| | Mild | 0 | 12 | 0 | 100% | |
| | Severe | 0 | 1 | 14 | 93% | |
| Valencia | Healthy | 3 | 0 | 0 | 100% | 97% |
| | Mild | 0 | 12 | 0 | 100% | |
| | Severe | 0 | 1 | 14 | 93% | |

CDA utilizing terpenoids between healthy, mild CTV infected, and severe CTV infected trees within each citrus cultivar. Different citrus cultivars were either healthy (mock-inoculated) or infected with mild or severe CTV strains.

That said, some overall trends due to CTV strain classifications were consistent across all cultivars. For soluble phenolics, overall levels in severe CTV strain-infected plants were lower than health controls. This could be due to severe strains counteracting responses by the host plants to reduce soluble phenolic levels, perhaps to block the formation of cell walls, as many phenolics are precursors to polyphenols, like lignin and tannins, that comprise cell walls. Severe CTV strains also could likely induce thicker cell walls, which in turn would remove some of the phenolics. Finally, reductions in phenolics could be due to a shift in photosynthate away from phenolics and towards amino acids and sugars.

By contrast, for total terpenoids there were significant differences between citrus cultivars. However, CTV infection status did not affect overall terpenoid levels. Because CTV infections seem to have minimal effects on host terpenoid levels, it would seem that baseline level measurements between tolerant and susceptible cultivars would be more meaningful, thus supporting the work by Guarino et al. [20].

For amino acids, differences in overall levels occurred due to both cultivar and infection status. Notably, severe CTV strain infections induced the greatest accumulation of amino acids, mild CTV strains resulted in an intermediate increase of amino acids, and healthy trees possessed the lowest amino acid levels. Increases in free amino acid content in infected trees may be the result of substantial amounts of amino acids being utilized for virus replication and assembly, effectively creating a metabolic "sink", as has been described for other plant-virus pathosystems [36]. Additionally, amino acids are precursors to many defense-related compounds (including hormones, proteins, and phenolics), so accumulation may be due to an increase of these compounds. Contrarily, the sugars analyzed in this study (fructose, sucrose, and glucose) differed due to cultivar, but not due to CTV infection status. Because sugars are associated with metabolomic sinks, it remains unresolved whether or not amino acid

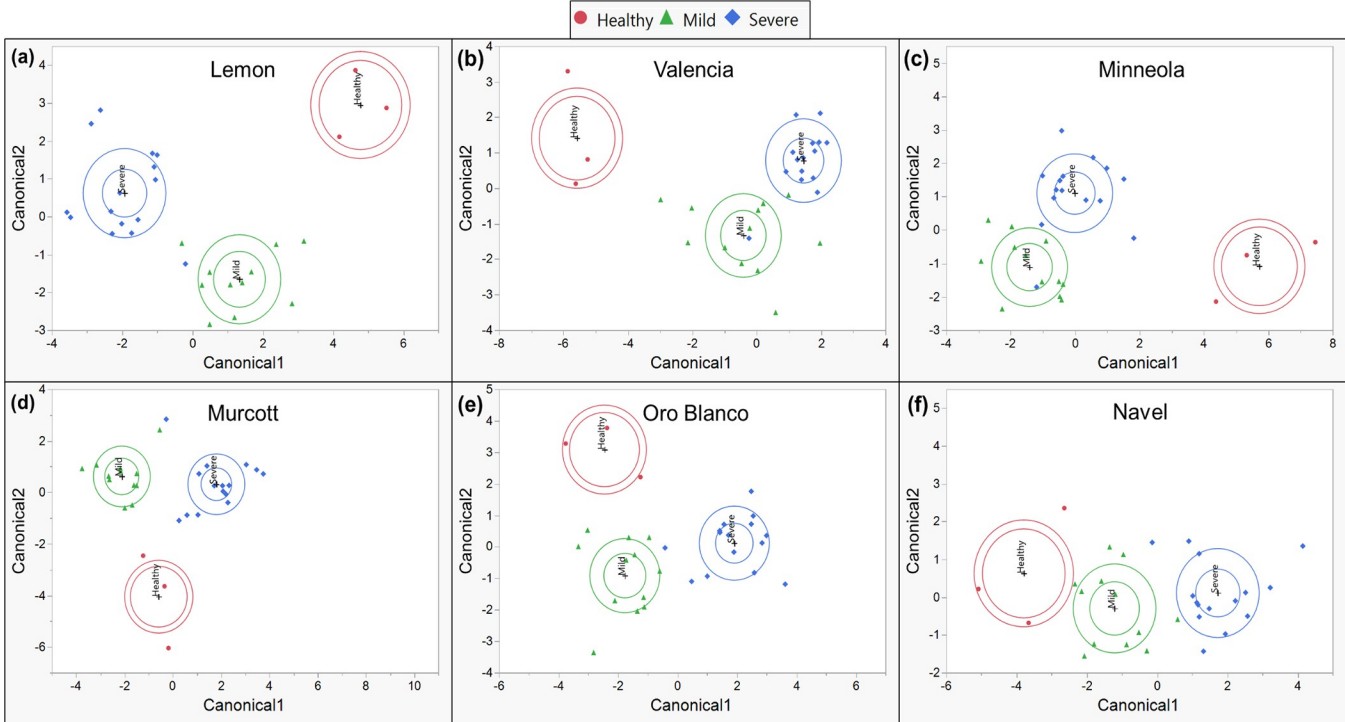

**Fig 10. Canonical plots using 10 most variable individual amino acids.** Canonical plots of different plants displaying mild (green-triangle), severe (blue-diamond), or no (red-circle) symptoms after CTV- or mock-inoculation, within each citrus cultivar. Canonicals were generated by stepwise discriminant analysis utilizing the 10 most variable amino acids between differentially infected plants of each cultivar. Outer ellipses are estimated 50% of the samples within a cultivar, and inner ellipses show the 95% confidence region of cultivar means.

modulation by CTV in citrus is due to its creation of a resource "sink" and should be further investigated. Overall, these findings were unexpected, and at times, patterns suggest potential trade-offs between primary metabolites (namely amino acids) at the expense of other secondary metabolites, especially phenolics. The mechanisms responsible for these metabolic shifts remain unknown and warrant future study.

Despite the basic findings about shifts in host plant chemistry, CDA utilization revealed larger trends that traditional statistics missed. Utilization of CDA with phenolics, terpenoids, and amino acids showed that just the ten most variable metabolites of each class were able to differentiate healthy citrus from CTV-infected trees and citrus infected with mild CTV from citrus infected with severe CTV. Of these, phenolic profiles were observed to be the most accurate in distinguishing infection status within cultivar, ranging from 97–100%. This could be, in part, due to the high number of different phenolic molecular species that were available for use in CDA, which is largely attributable to their structural diversity and ability to be simultaneously detected.

This study shows the potential that CDA has at being a useful tool for discernment of trends and patterns in biochemical shifts that are otherwise not readily observable. Additionally, CDA can potentially be used to identify infected trees and ascertain whether infections are caused by severe or mild CTV strains. Of course, this assumes consistency in standardizing samples is possible, including using the same cultivar in similar growth conditions. Likewise, a more complete understanding about differences in the relative severity of CTV strains is still required, as presumably different strains could, in the end, result in similar symptomology and similar host responses [14]. Further research is needed to verify if CDA is consistently accurate

**Table 5. CDA utilizing amino acids.**

| Cultivar | CTV Infection Status | Predicted Count | | | Accuracy | |
|---|---|---|---|---|---|---|
| | | Healthy | Mild | Severe | Match % | Overall % |
| Lisbon | Healthy | 3 | 0 | 0 | 100% | 97% |
| | Mild | 0 | 11 | 0 | 100% | |
| | Severe | 0 | 1 | 14 | 93% | |
| Minneola | Healthy | 3 | 0 | 0 | 100% | 93% |
| | Mild | 0 | 12 | 0 | 100% | |
| | Severe | 0 | 2 | 13 | 87% | |
| Murcott | Healthy | 3 | 0 | 0 | 100% | 97% |
| | Mild | 0 | 12 | 0 | 100% | |
| | Severe | 0 | 1 | 14 | 93% | |
| Navel | Healthy | 3 | 0 | 0 | 100% | 93% |
| | Mild | 0 | 11 | 1 | 92% | |
| | Severe | 0 | 1 | 14 | 93% | |
| Oro Blanco | Healthy | 3 | 0 | 0 | 100% | 97% |
| | Mild | 0 | 12 | 0 | 100% | |
| | Severe | 0 | 1 | 14 | 93% | |
| Valencia | Healthy | 3 | 0 | 0 | 100% | 90% |
| | Mild | 0 | 10 | 2 | 83% | |
| | Severe | 0 | 1 | 14 | 93% | |

CDA utilizing amino acids between healthy, mild CTV infected, and severe CTV infected trees within each citrus cultivar. Different citrus cultivars were either healthy (mock-inoculated) or infected with mild or severe CTV strains.

over a larger variety of conditions, but it represents a potentially powerful tool for management of CTV in commercial orchards.

## Supporting information

**S1 Appendix. Supplementary tables providing summaries of additional ANOVA statistics.** Table A represents statistics for individual phenolics, Table B represents statistics for individual terpenoids, and Table C represents statistics for individual amino acids. (PDF)

## Acknowledgments

The authors thank Nalong Mekdara and Robert Deborde for their assistance in this research. Mention of trade names or commercial products in this publication is solely for the purpose of providing specific information and does not imply recommendation or endorsement by the U.S. Department of Agriculture. USDA is an equal opportunity provider and employer.

## Author Contributions

**Conceptualization:** Christopher M. Wallis, Subhas Hajeri, Raymond Yokomi.

**Data curation:** Christopher M. Wallis, Zachary Gorman, Rachel Rattner, Subhas Hajeri, Raymond Yokomi.

**Formal analysis:** Christopher M. Wallis, Zachary Gorman, Subhas Hajeri, Raymond Yokomi.

**Funding acquisition:** Christopher M. Wallis, Subhas Hajeri, Raymond Yokomi.

**Investigation:** Christopher M. Wallis, Rachel Rattner, Subhas Hajeri, Raymond Yokomi.

**Methodology:** Christopher M. Wallis, Rachel Rattner, Subhas Hajeri, Raymond Yokomi.

**Project administration:** Christopher M. Wallis, Raymond Yokomi.

**Resources:** Christopher M. Wallis, Raymond Yokomi.

**Software:** Christopher M. Wallis, Zachary Gorman, Raymond Yokomi.

**Supervision:** Christopher M. Wallis, Raymond Yokomi.

**Validation:** Christopher M. Wallis.

**Visualization:** Christopher M. Wallis.

**Writing – original draft:** Christopher M. Wallis, Zachary Gorman.

**Writing – review & editing:** Christopher M. Wallis, Zachary Gorman, Rachel Rattner, Raymond Yokomi.

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
