## [Decision Letter · Decision Letter 0]

15 Mar 2022

PONE-D-22-03106Amino acid, sugar, phenolic, and terpenoid profiles are capable of distinguishing Citrus tristeza virus infection status in grapefruit, lemon, mandarin, and sweet orange leavesPLOS ONE

Dear Dr. Wallis,

Thank you for submitting your manuscript to PLOS ONE. After careful consideration, we feel that it has merit but does not fully meet PLOS ONE’s publication criteria as it currently stands. Therefore, we invite you to submit a revised version of the manuscript that addresses the points raised during the review process.

Meaningfulness of CDA analysis is rather questionable, according to reviewers' reports. Therefore, deepening the Discussion section in the direction of clarifying these results and conclusions built out of them should be in the main authors' focus for manuscript revision. Please be aware of additional comments provided in the attached file by Reviewer #1.

We look forward to receiving your revised manuscript.

Kind regards,

Branislav T. Šiler, Ph.D.

Academic Editor

PLOS ONE

Journal Requirements:

Reviewers' comments:

Reviewer's Responses to Questions

**Comments to the Author**

1. Is the manuscript technically sound, and do the data support the conclusions?

Reviewer #1: Yes

Reviewer #2: No

2. Has the statistical analysis been performed appropriately and rigorously? 

Reviewer #1: Yes

Reviewer #2: I Don't Know

3. Have the authors made all data underlying the findings in their manuscript fully available?

Reviewer #1: Yes

Reviewer #2: No

4. Is the manuscript presented in an intelligible fashion and written in standard English?

Reviewer #1: Yes

Reviewer #2: Yes

5. Review Comments to the Author

Reviewer #1: This manuscript Aminoacid,sugar,phenolic,andterpenoidprofilesarecapableofdistinguishingCitrustristezavirusinfectionstatusingrapefruit,lemon, mandarin,and sweet orange leaves” is well organized and provides an assessment of changes the level of amino acid, sugar, phenolic, and terpenoid in healthy and CTV-infected grapefruit, lemon, mandarin, and sweet orange cultivars. The results could improve understanding about citrus-CTV interactions and biochemistry in CTV infected plant.The study has shown novelty and usefulness of the data for distinguishing infection by evere and mild CTV in infected citrus trees, thus this MS could be published.

Reviewer #2: The manuscript "Amino acid, sugar, phenolic, and terpenoid profiles are capable of distinguishing Citrus tristeza virus infection status in grapefruit, lemon, mandarin, and sweet orange leaves" provides an interesting overview of several metabolites in different citrus species.

The paper is quite intellegible. However it is not publisheable in the present version.

Taking into account that the finding of differences among the metabolite profiles in different citrus species and cultivars is well known and usually reported in the literature, those data are not new.

Authors claim that using their data it could be possible to distinguish the infection status in plants infected by CTV. In my opinion this affirmation is not correct, taking into consideration the presented results, as the only significant differences as a function of the infection status are in the levels of total phenolics between healthy and severe status in Lisbon cultivar.

Also authors wrote " Oro Blanco had the least amino acids of any citrus cultivar tested. CTV status also affected total amino acids levels, with citrus infected with severe CTV strains possessing greater levels of total amino acids than citrus infected with mild CTV strains or healthy trees (F2, 161 = 17.708; P < 0.001), but in the relative figure .n5) the different infection status presented the same letter (a) and it means no statistical significant differences.

I found difficult to understand how authors made CDA analysis, when no significant differences are present in the metabolite levels.

For this reason the discussion is at present too speculative and should be written again after presenting the results in correct way.

The levels of each metabolite in each citrus species/cultivar should be provided to control the reliability of the statistical analysis at present onbly for review purpose and then loaded in a public repository.

In Fig 2b and 2c and Fig. 5: There are no letters or simbols showing significant differences among the citrus species and cultivars.

6. PLOS authors have the option to publish the peer review history of their article (what does this mean?). If published, this will include your full peer review and any attached files.

Reviewer #1: No

Reviewer #2: No

---

## [Author Response · Author response to Decision Letter 0]

5 Apr 2022

Amino acid, sugar, phenolic, and terpenoid profiles are capable of distinguishing Citrus tristeza virus infection status in grapefruit, lemon, mandarin, and sweet orange leaves

Journal Requirements: Return Review by April 29, 2022

Author Response: The manuscript has been edited to match journal requirements.

Author Response: The Supporting Information files have been re-named, captions added at the end of the manuscript, and in-text citations adjusted to meet journal guidelines.

Author Response: The reference list has been edited to ensure it is complete and correct.

Editor Comment:

Meaningfulness of CDA analysis is rather questionable, according to reviewers' reports. Therefore, deepening the Discussion section in the direction of clarifying these results and conclusions built out of them should be in the main authors' focus for manuscript revision. Please be aware of additional comments provided in the attached file by Reviewer #1.

Author Response: Overall, it appeared some confusion over the nature of the CDA that resulted in the comments question its meaningfulness. Namely, the figures, which represent summed total amounts of entire compound classes, where presumed to be used in the CDA. This was not the case, and instead individual compound values were utilized, which were consistently suggested to be significantly different due to infection status by MANOVA. It is in this light that the CDA should be viewed, and not trying to make unequal comparisons with the Figures. The text was edited throughout to make this point clearer. A new Figure (Fig 5), a heatmap showing differences in individual compounds, was added. Moving the Tables from the Supplementary file to the manuscript could be another option to clarify and support the CDAs, but that makes the manuscript unwieldly even further. That said, we the authors will be fine if the editor wants to do such.

Reviewer 1 Comments:

This manuscript “Amino acid, sugar, phenolic, and terpenoid profiles are capable of distinguishing Citrus tristeza virus infection status in grapefruit, lemon, mandarin, and sweet orange leaves” is well organized and provides an assessment of changes the level of amino acid, sugar, phenolic, and terpenoid in healthy and CTV-infected grapefruit, lemon, mandarin, and sweet orange cultivars. The results could improve understanding about citrus-CTV interactions and biochemistry in CTV infected plant. The study has shown novelty and usefulness of the data for distinguishing infection by severe and mild CTV in infected citrus trees, thus this MS could be published.

The title should be changed like: “Amino acid, sugar, phenolic, and terpenoid profiles are capable of distinguishing Citrus tristeza virus infection status in citrus cultivars grapefruit, lemon, mandarin, and sweet orange”.

Author Response: The title was changed as suggested.

Line 24-30: In abstract portion ( However, canonical discriminant analysis (CDA) utilizing profiles of individual amino acids, terpenoids, or phenolics was able to correctly match leaf samples to specific citrus varieties and identify their infection status with good accuracy. This analysis reliably distinguished citrus infected with mild CTV strains from those infected with severe CTV strains. Collectively, this study reveals biochemical patterns associated with severity of CTV infections that can potentially be utilized to help identify in-field CTV infections of economic relevance.)- Should be more clear

Author Response: These sentences were revised to be clearer.

Line 55-76: Introduction portion (Paragraphs 3 and 4) should be merged and to be shorten. Symptomatology should not be so highlighted as correlation with the symptom severity and metabolic changes was not shown clearly in this paper 

Author Response: The two mentioned paragraphs have been combined and greatly reduced for brevity as suggested.

References cited in the text is not accordingly the journal pattern-should be corrected (numbering?)

Bibliography list of references should be accordingly the PLOS ONE pattern. Consult any recent reprint of PLOS ONE and correct it accordingly.

Author Response: The references were re-formatted to match journal guidelines.

Line 441: “Ultimately, this study provides unique insight into the biochemical mechanisms associated with CTV, establishing distinct metabolic patterns associated with the differential 443 severity of CTV strain?.--biochemical mechanisms- I feel this terms are not required here. 

Author Response: This sentence was removed as requested.

Line 443: “This study shows CDAs are a useful tool for discernment of trends and 444 patterns in biochemical shifts that are otherwise not readily observable.” CDAs could be useful… as more standardization is needed. It has tremendous potentiality to distinguishing the infection caused by severe and mild strain as the antisera, developed as specific (called) to severe and mild CTV, could not always work. As in introduction it was mentioned “However, 59 a recent study observing MCA13+ and MCA13 negative (MCA13-) strains infecting trees in the 60 same orchards did not observe differences in symptoms or yields (Yokomi et al. 2020).

Author Response: This point was made by adding a new sentence to the final paragraph, calling for the need to better describe CTV strains and the severity of diseases they cause. A new Fig. 1 was added to show the difference between mild and severe CTV symptomology in Washington Navel. Severe CTV causes debilitating stem pitting in sensitive orange and grapefruit cultivars which leads to stunting, brittle branches, and yield reduction. 

Reviewer 2 Comments:

The manuscript "Amino acid, sugar, phenolic, and terpenoid profiles are capable of distinguishing Citrus tristeza virus infection status in grapefruit, lemon, mandarin, and sweet orange leaves" provides an interesting overview of several metabolites in different citrus species.

The paper is quite intelligible. However it is not publishable in the present version.

Taking into account that the finding of differences among the metabolite profiles in different citrus species and cultivars is well known and usually reported in the literature, those data are not new.

Author Response: There were multiple significant effects due to CTV strain found when individual compounds were analyzed, as indicated by the significant MANOVA (and follow-up ANOVA) tests for phenolics, terpenoids, and amino acids, which is clearly stated within the text but admittedly not represented within the figures examining the summed total of this compound classes. Examining the Supplementary Tables makes this clear. We also added a new Figure (Fig 5) to further show these differences. These findings are therefore a solid addition to previous reports.

Authors claim that using their data it could be possible to distinguish the infection status in plants infected by CTV. In my opinion this affirmation is not correct, taking into consideration the presented results, as the only significant differences as a function of the infection status are in the levels of total phenolics between healthy and severe status in Lisbon cultivar.

Author Response: The reviewer based this and other comments mostly on examining the Figures, but missed the MANOVA (and follow-up ANOVA) analyses within the text examining individual phenolic, terpenoid, and amino acid compounds, which did significantly vary due to infections overall. A new Figure (Figure 5) also shows these observations. The individual compounds were utilized in the CDA analyses, not the summed total compound levels that comprised the Figures 4, 6-7, and therefore conclusions were able to be reached otherwise. 

Also authors wrote " Oro Blanco had the least amino acids of any citrus cultivar tested. CTV status also affected total amino acids levels, with citrus infected with severe CTV strains possessing greater levels of total amino acids than citrus infected with mild CTV strains or healthy trees (F2, 161 = 17.708; P < 0.001), but in the relative figure .n5) the different infection status presented the same letter (a) and it means no statistical significant differences.

Author Response: This was a mis-reading of the stated text, as the sentences were not meant to flow into each other. A new paragraph was inserted and a supporting clause to clarify the results of the ANOVA more accurately, that is, when means from healthy, mild-, or severe-strain infected plants from ALL of the cultivars was considered, there was a significant difference present. This was not represented by the letters present in Figure 7 (formerly Figure 5), which was only for within-cultivar differences (not for differences amongst all cultivars). 

I found difficult to understand how authors made CDA analysis, when no significant differences are present in the metabolite levels. For this reason the discussion is at present too speculative and should be written again after presenting the results in correct way.

Author Response: There were multiple, consistent differences present in individual metabolite levels as suggested by the significant MANOVAs and follow-up ANOVAs as described in the text. The heatmap in Fig 5 shows these observations, and it is supported by the Supplementary material. A quick look at the Figures would be misleading as those figures focused on summed amounts (total phenolic, terpenoid, or amino acid levels), and not individual compound levels (which are present in the Supplementary file). The CDA analyses was NOT based on the summed compound data from the figures, rather on the individual compound data (as covered in the various Tables), and therefore should be interpreted in that light only.

The levels of each metabolite in each citrus species/cultivar should be provided to control the reliability of the statistical analysis at present only for review purpose and then loaded in a public repository.

Author Response: The data have already been uploaded to a depository, the Ag Data Commons, which is run by the United States National Agricultural Library and is freely accessible. There is an embargo until early April though, and therefore the uploaded file will be provided with this revision for the editor’s and reviewer’s use.

In Fig 2b and 2c and Fig. 5: There are no letters or symbols showing significant differences among the citrus species and cultivars.

Author Response: Indeed, the letters represent differences of the virus for each individual cultivar, not overall differences amongst cultivars or overall differences in infection status regardless of cultivar. The text provides this information instead.

---

## [Editor Report · Decision Letter 1]

13 Apr 2022

PONE-D-22-03106R1Amino acid, sugar, phenolic, and terpenoid profiles are capable of distinguishing Citrus tristeza virus infection status in grapefruit, lemon, mandarin, and sweet orange leavesPLOS ONE

Dear Dr. Wallis,

Thank you for submitting your manuscript to PLOS ONE. After careful consideration, we feel that it has merit but does not fully meet PLOS ONE’s publication criteria as it currently stands. Therefore, we invite you to submit a revised version of the manuscript that addresses the points raised during the review process.

In their response, the authors claimed they changed the main title into "Amino acid, sugar, phenolic, and terpenoid profiles are capable of distinguishing Citrus tristeza virus infection status in citrus cultivars grapefruit, lemon, mandarin, and sweet orange” as suggested by Reviewer #1. However, they missed to do so.

Please add a colon after "cultivars".

We look forward to receiving your revised manuscript.

Kind regards,

Branislav T. Šiler, Ph.D.

Academic Editor

PLOS ONE
---

## [Author Response · Author response to Decision Letter 1]

13 Apr 2022

Editor Comment:

In their response, the authors claimed they changed the main title into "Amino acid, sugar, phenolic, and terpenoid profiles are capable of distinguishing Citrus tristeza virus infection status in citrus cultivars grapefruit, lemon, mandarin, and sweet orange” as suggested by Reviewer #1. However, they missed to do so.

Please add a colon after "cultivars".

Author Response: We apologize for the oversight. The title has been changed as suggested, including adding the colon after “cultivars”.

---

## [Editor Report · Decision Letter 2]

26 Apr 2022

Amino acid, sugar, phenolic, and terpenoid profiles are capable of distinguishing Citrus tristeza virus infection status in citrus cultivars: grapefruit, lemon, mandarin, and sweet orange

PONE-D-22-03106R2

Dear Dr. Wallis,

We’re pleased to inform you that your manuscript has been judged scientifically suitable for publication and will be formally accepted for publication once it meets all outstanding technical requirements.

Kind regards,

Branislav T. Šiler, Ph.D.

Academic Editor

PLOS ONE
---

## [Editor Report · Acceptance letter]

29 Apr 2022

PONE-D-22-03106R2 

Amino acid, sugar, phenolic, and terpenoid profiles are capable of distinguishing *Citrus tristeza virus* infection status in citrus cultivars: grapefruit, lemon, mandarin, and sweet orange 

Dear Dr. Wallis:

I'm pleased to inform you that your manuscript has been deemed suitable for publication in PLOS ONE. Congratulations! Your manuscript is now with our production department. 

Kind regards, 

on behalf of

Dr. Branislav T. Šiler 

Academic Editor

PLOS ONE